# PERSONALIZED PROMPT TUNING FOR UNSUPERVISED FEDERATED LEARNING

## ABSTRACT

Federated learning facilitates collaborative model training across multiple distributed clients without requiring data sharing. However, conventional federated methods struggle with classification tasks in an unsupervised paradigm due to the absence of category knowledge. Recently, CLIP, a prominent visual language model, has demonstrated impressive results, particularly its remarkable zero-shot classification ability, which alleviates the dependence on labeled data. In this paper, we first explore a new realistic problem, unsupervised federated learning using CLIP, where clients with unlabeled heterogeneous data collaborate to enhance global performance. To address this problem, we propose FedPP, a method that incorporates a cooperative pseudo-label selection strategy and a partial prompt aggregation protocol. Our selection strategy ensures that all classes are trained in a balanced manner through global pseudo-label allocation. Concurrently, the aggregation protocol divides parameters into aggregated and retained components to optimize global performance while supporting local personalization. Extensive experiments across six datasets with various types of heterogeneity demonstrate the effectiveness of FedPP. Our code is available in the **supplementary materials**.

## 1 INTRODUCTION

Federated Learning (McMahan et al., 2017) is a distributed machine learning framework that enables decentralized collaboration among clients without sharing their local training data. For instance, in multi-institutional healthcare collaborations, participating clients (e.g., medical institutions and hospitals) can collaboratively train powerful models without leaking patient information (Liu et al., 2021). The conventional federated algorithms typically operate within a system composed of clients with labeled data, following an iterative process of local training at the client level and global aggregation at the server level (McMahan et al., 2017; Sheller et al., 2020; Li et al., 2020; Luo et al., 2021; Zhang et al., 2023b; Guo et al., 2024). Additionally, researchers have begun to apply federated techniques in unsupervised learning settings (Zhang et al., 2023a; Tian et al., 2024; Nardi et al., 2024; Lu et al., 2022; Lubana et al., 2022). Existing unsupervised federated methods primarily concentrate on leveraging self-supervised algorithms for representation learning tasks and clustering tasks. However, the absence of category knowledge presents a significant challenge for classification tasks involving unlabeled datasets in federated learning.

Recently, pre-trained vision-language models (VLMs), such as CLIP (Radford et al., 2021), have shown remarkable representation and generalization capabilities across various downstream tasks, such as image classification (Cho et al., 2023; Kan et al., 2023; Liang et al., 2024), semantic segmentation (Wang et al., 2022; Xu et al., 2022; Liang et al., 2023), and object detection (Esmaeilpour et al., 2022; Chen et al., 2024). Benefiting from pre-training on large-scale image-text pairs, CLIP has powerful zero-shot classification capability. This is achieved by utilizing the textual encoder to generate classifier weights from a simple prompt, such as "a photo of a [CLASS]". The impressive zero-shot prediction capability of CLIP eliminates the reliance on labeled data, opening new avenues for unsupervised federated learning.

In this paper, we investigate a novel and realistic problem, unsupervised federated learning with CLIP, where clients leverage unlabeled data to collaborate on complex problems such as image classification. In this setting, each client is equipped with an identically initialized pre-trained CLIP and the names of target categories. Similar to standard federated learning frameworks, clients periodi-

cally upload their non-data knowledge (e.g., statistical information, model parameters) to the server for aggregation and download global knowledge in return. However, due to the nature of unsupervised learning and the federated paradigm, label skews among clients can easily lead to negative transfer and aggregation failure.

To address this problem, we propose an unsupervised federated method called FedPP. FedPP incorporates two key components, the cooperative pseudo labels selection (CPS) strategy and the partial prompt aggregation (PPA) protocol. The CPS strategy focuses on selecting reliable pseudo labels that enhance both local training and global aggregation for each client. Specifically, to improve performance across all categories, we globally select pseudo labels by category and distribute them to participating clients based on their estimated pseudo label distribution. This approach mitigates the category bias caused by the inherent bias of CLIP, improving the accuracy of the pseudo labels.

Bisides, the PPA protocol uploads only visual prompts to the server for aggregation, while keeping textual prompts retained locally to enhance personalization and avoid aggregation conflicts. The reason for only choosing visual prompts for aggregation is that visual prompts often learn general representations of the image domain while textual prompts tend to learn category-related information which may introduce conflicts due to label skews. Consequently, global performance is improved through the aggregated visual prompts, while clients benefit from personalized textual prompts better suited to their data distribution. Experimental results across standard federated prompt learning benchmarks with both Dirichlet-based and quantity-based label skews demonstrate the effectiveness of the proposed FedPP method. Our main contributions are summarized as follows:

- We introduce unsupervised federated learning with CLIP, a novel and realistic problem where clients with unlabeled data collaborate to improve model performance.
- To address the unsupervised federated learning problem, we propose FedPP, which comprises two key components: cooperative pseudo labels selection, which ensures balanced training across all classes through global pseudo-label allocation, and partial prompt aggregation, which separates parameters into aggregated and retained components.
- Extensive experiments across six datasets with different types of heterogeneity demonstrate the effectiveness of FedPP.

## 2 RELATED WORKS

### 2.1 VISION-LANGUAGE MODELS

Recently, vision-language models (VLMs) such as CLIP (Radford et al., 2021), ALBEF (Li et al., 2021), BLIP (Li et al., 2022a), and Flamingo (Alayrac et al., 2022), pre-trained on large-scale image-text data, have achieved significant success in zero-shot learning (Lu et al., 2019; Wang et al., 2024b; Liang et al., 2024; Wang et al., 2023; Kim et al., 2021; Zhang et al., 2024b), in-context learning (Zhou et al., 2024; Doveh et al., 2024), open-world segmentation (Tang et al., 2024; Ha & Song, 2022), and open-world detection (Wang et al., 2024a). Furthermore, the performance of VLMs can be enhanced through fine-tuning with annotated data from downstream datasets (Zhou et al., 2022b;a; Udandarao et al., 2023). For instance, CoOp (Zhou et al., 2022b) learns textual prompts for downstream tasks via back-propagation on few-shot datasets, while CoCoOp (Zhou et al., 2022a) incorporates visual information into textual prompts for regularization, improving base-to-new generalization performance. Additionally, CLIP-Adapter (Gao et al., 2024) proposes learning both a visual and a textual adapter to refine the original representations of vision-language models. In this paper, we primarily focus on the performance of CLIP as a representative VLM in downstream tasks.

### 2.2 FEDERATED LEARNING WITH VISION-LANGUAGE MODELS

Federated Learning (FL) (McMahan et al., 2017) has emerged as a pivotal paradigm for decentralized training of machine learning models on heterogeneous data, preserving data privacy and reducing data transfer overhead (Qu et al., 2022; Li et al., 2023b; Guo et al., 2024; Shi et al., 2023; 2024). Recently, the fine-tuning of Vision-Language Models (VLMs) has been extended to the federated framework to alleviate the computational load on individual devices while addressing challenges in federated learning, such as subpar performance and robustness in cross-domain scenarios, as well as non-IID data distributions among clients (Qiu et al., 2023; Li & Wang, 2024; Halbe et al., 2023;

Su et al., 2024; Yang et al., 2023). For example, FedCLIP (Lu et al., 2023) directly extends the standard fine-tuning of CLIP to the federated setting to achieve strong performance and personalization. PromptFL (Guo et al., 2023b) introduces a federated learning framework for prompt learning, enabling participants to collaboratively learn a common prompt vector. pFedprompt (Guo et al., 2023a) combines a federated prompt learning scheme with personalized spatial visual features. Additionally, pFedPG (Yang et al., 2023) generates personalized prompts for each client based on their visual prompts to better align with their data distribution. FedOPT (Li et al., 2024) utilizes knowledge from both personal and global textual prompts for prediction through non-uniform optimal transport. However, previous methods have primarily focused on supervised federated learning with VLMs. In this paper, we further explore leveraging VLMs for unsupervised federated learning, capitalizing on their zero-shot capabilities.

### 2.3 Unsupervised Learning for vision-language models

In real-world applications, high annotation costs are often required to ensure that each data source has labeled data. This necessity drives us to develop effective methods for utilizing unlabeled data in downstream tasks. Pseudo-labeling strategy (Huang et al., 2022; Menghini et al., 2023; Zhang et al., 2024a; Jia et al., 2024; Tanwisuth et al., 2023) and entropy minimization (Liang et al., 2024) are widely studied. For instance, UEO (Liang et al., 2024) leverages sample-level confidence to minimize the conditional entropy of confident instances while maximizing the marginal entropy of less confident ones. POUF (Tanwisuth et al., 2023) fine-tunes the model or prompt by aligning the discrete distributions derived from the prompts and unlabeled target data. UPL (Huang et al., 2022) and FPL (Menghini et al., 2023) select an equal number of pseudo-labels for each category, while CPL (Zhang et al., 2024a) generates multiple pseudo labels for each sample to enhance labeling accuracy. However, existing pseudo labeling strategies have proven challenging to apply directly to the zero-shot predictions of VLMs in federated settings (Huang et al., 2022). In this paper, we propose a novel cooperative pseudo labels selection strategy to mitigate the category bias introduced by the inherent biases of pre-trained VLMs and significantly improve the accuracy of pseudo labels.

## 3 Method

In this section, we present a detailed overview of our problem and proposed method. First, in Sec. 3.1, we review foundational concepts related to CLIP (Radford et al., 2021) and prompt tuning methods. Next, Sec. 3.2 provides an overview of our unsupervised heterogeneous federated problem for Vision Language Models (VLMs). We then introduce our proposed method, FedPP, which is built upon two key strategies: cooperative pseudo-label selection discussed in Sec. 3.3, and partial prompt aggregation covered in Sec. 3.4.

### 3.1 Preliminaries

In this paper, we adopt CLIP (Radford et al., 2021) as the foundational model. CLIP utilizes a dual-branch architecture consisting of an image encoder, $F^v(\cdot)$, and a text encoder, $F^t(\cdot)$, with each encoder processing data from its respective modality. For zero-shot predictions in downstream tasks, CLIP utilizes a human-designed prompt (e.g., "a photo of a [CLASS]") for each class. Take a C-way classification task as an example, textual embeddings of all classes $\{f_c^t\}_{c=1}^C$ and the embedding of the test image $f^v(x)$ are derived from the text and image encoders, respectively. The probability that image x belongs to the $c$-th category is calculated after applying the softmax operation:

$$p_c(x) = \frac{\exp(\text{sim}(f^v(x), f_c^t)/\tau)}{\sum_{j=1}^C \exp(\text{sim}(f^v(x), f_j^t)/\tau)}, \tag{1}$$

where $\tau$ is a temperature parameter. To enhance performance in downstream tasks, prompt tuning is widely adopted as a parameter-efficient fine-tuning method. This involves introducing additional learnable textual tokens $P^t$ and visual tokens $P^v$ (Zhou et al., 2022b; Jia et al., 2022; Xing et al., 2023) (referred to as textual/visual prompts) into the corresponding encoders, thereby optimizing the original CLIP model for specific applications.

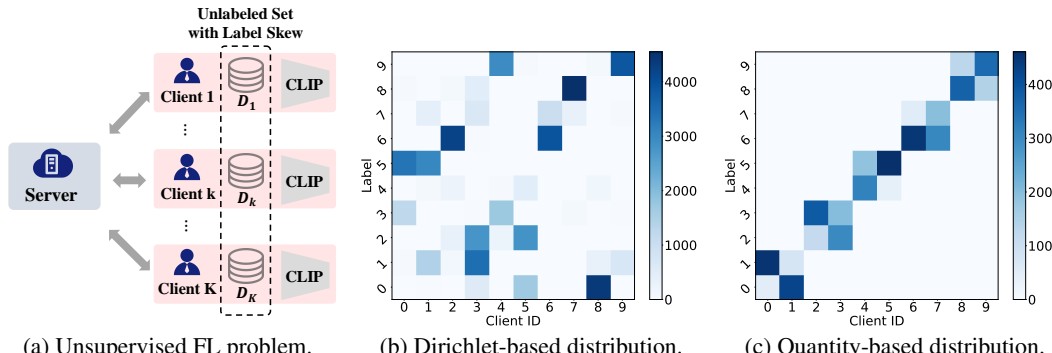

(a) Unsupervised FL problem. (b) Dirichlet-based distribution. (c) Quantity-based distribution.

Figure 1: Illustration of unsupervised federated learning problem. (a) displays the unsupervised federated learning problem, where each client possesses unlabeled data and an identically initialized pre-trained vision-language model (VLM). (b) and (c) depict label skews of the data distribution among clients.

## 3.2 UNSUPERVISED HETEROGENEOUS FEDERATED PROBLEM FOR VLMS

The powerful zero-shot classification capabilities of CLIP alleviate the requirement of labeled data in federated learning scenarios. We investigate an unsupervised heterogeneous federated learning problem involving a central server and $K$ remote clients, as illustrated in Figure 1 (a). Each client possesses an unlabeled local dataset $D_k$ with a capacity of $n_k$ and an identically initialized CLIP. Similar to standard federated learning frameworks, clients periodically upload their non-data knowledge (e.g., statistical information, model parameters) to the server for aggregation and download global knowledge in return. A challenge of the problem is heterogeneity, characterized by label skew in the unlabeled datasets across clients. Specifically, the union of all clients' datasets $\bigcup_k D_k$ encompasses data from all categories, while $D_k$ may only represent a subset of these categories. Additionally, the number of samples for each category can vary among clients. Figure 1 (b) and (c) illustrate client data distribution under Dirichlet-based and quantity-based label skews, respectively. Further details on heterogeneity can be found in Sec. 4.1. Label skews among clients' unlabeled datasets can lead to negative transfer and pose significant risks to federated aggregation.

## 3.3 COOPERATIVE PSEUDO LABELS SELECTION

To tackle the challenges associated with unsupervised heterogeneous federated learning, we propose an effective solution called FedPP. This approach integrates two core strategies: a cooperative pseudo label selection strategy and partial prompt aggregation. The illustration of our method is shown in Figure 2, and the pseudocode of the algorithm can be found in the appendix A.1.

Given the heterogeneity of unlabeled datasets among clients, directly applying conventional pseudo label selection methods can lead to an imbalanced global distribution and low-quality pseudo labels. To address this, we introduce a cooperative pseudo label selection strategy that ensures the union of pseudo labels across all clients maintains a uniform distribution across categories. Simultaneously, the pseudo labels for each client are tailored to fit their specific local data distribution.

To facilitate class assignments on the server, clients first estimate the local label distribution. For client $k$, we filter the reliable samples from the unlabeled set $D_k$ based on the confidence and entropy of model predictions to construct the estimated set $D_k^{est}$ as follows:

$$D_k^{est} = \{(x, \hat{y}) | \max_c p_c(x) > p_{\frac{1}{2}} , \ Ent(p_c(x)) < H_{\frac{1}{2}}\}, \tag{2}$$

where $\hat{y} = \arg\max_c p_c(x)$ represents the predicted label, and $Ent(\cdot)$ denotes the Shannon Entropy operator. Here, $p_{\frac{1}{2}}$ and $H_{\frac{1}{2}}$ are the medians of confidence and entropy within the alternative set, respectively. We then compute statistics on the pseudo labels by category, yielding the estimated distribution $U_k^{est}$, a $C$-dimensional vector where the $c$-th element $u_{k,c} = \sum_i \mathbb{1}(\hat{y}_i = c)$ indicates the number of samples associated with class $c$ according to the pseudo labels.

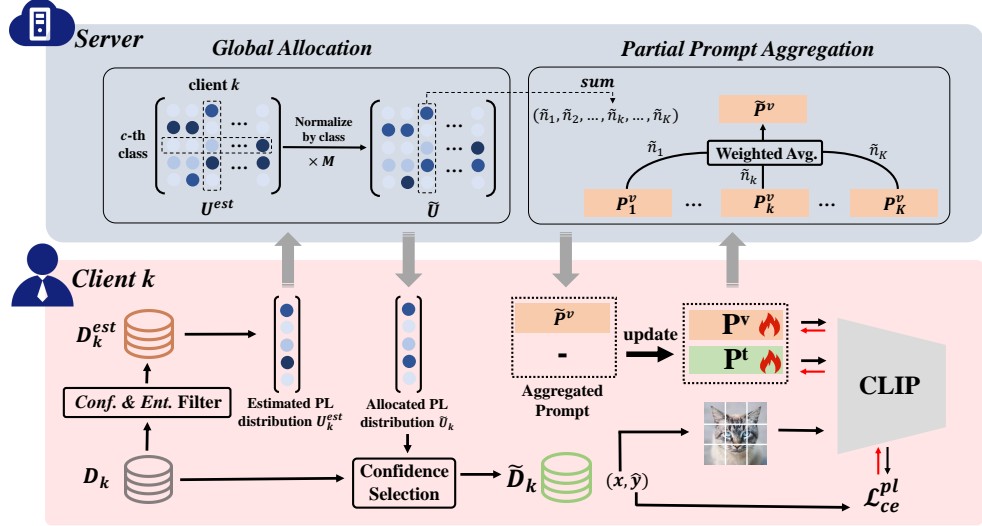

Figure 2: **The overview of our FedPP.** For pseudo labels generation, FedPP begins by filtering the reliable samples to estimate the local label distribution, which is aggregated as global estimated label distribution in the server. Then, the server globally selects $M$ pseudo labels for each category and allocates them to clients based on local and global estimated distributions. To handle label skews during training, we aggregate only visual prompts on the server to enhance global performance because the differences in textual prompts are significantly greater than those found in visual prompts.

Subsequently, clients upload their estimated distribution $U_k^{est}$ to the server for the collaborative assignment. To ensure adequate training for all categories, the server globally selects $M$ pseudo labels for each category and allocates them to clients based on their estimated distributions, as follows:

$$\widetilde{U}_k = (\widetilde{u}_{k,1}, \widetilde{u}_{k,2}, ..., \widetilde{u}_{k,C}), \text{ where } \widetilde{u}_{k,c} = \left\lceil \frac{u_{k,c}}{\sum_i u_{i,c}} \cdot M \right\rceil \quad (3)$$

denotes the amount of training data of category $c$ assigned to client $k$.

Finally, client $k$ constructs the training set $\widetilde{D}_k$ based on its capacity $\widetilde{U}_k$. The $\widetilde{u}_{k,c}$ samples with the highest prediction probability for the $c$-th class in the original dataset $D_k$ are selected and added to the training set $\widetilde{D}_k$. As federated training progresses, we will periodically repeat these steps to generate, estimate, and assign pseudo-labels, thereby obtaining new high-quality training data $\widetilde{D}_k$. The cooperative pseudo labels selection strategy ensures uniform training across all categories globally, establishing a solid foundation for aggregation. Additionally, the assigned training data for each client is tailored based on the estimated distribution, enhancing the local model's consistency with its specific data distribution.

### 3.4 PARTIAL PROMPT AGGREGATION

For each client, we select both textual prompt $P_k^t$ and visual prompt $P_k^v$ as the optimization parameters to be trained using the local training set $\widetilde{D}_k$. Since each client's training set pertains to the same task, effective knowledge aggregation can enhance the overall framework's performance. A straightforward approach is to aggregate the prompts using a simple averaging operation. However, this method may lead to suboptimal or even detrimental performance. As for CLIP, the updates to the visual branch primarily enhance image representation knowl-

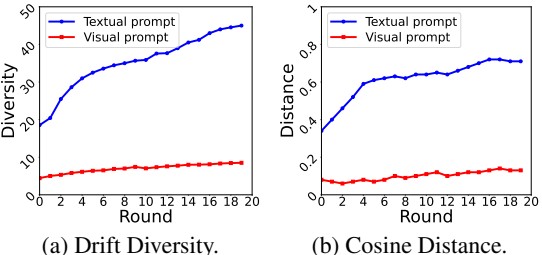

(a) Drift Diversity.      (b) Cosine Distance.

Figure 3: Drift diversity and cosine distance of prompts among clients during training in CIFAR10 (Krizhevsky et al., 2009) dataset. The differences observed in textual prompts are significantly greater than those found in visual prompts.

edge, while the textual branch focuses on determining the classification boundaries by leveraging category information. Consequently, under heterogeneous data distributions among clients, the visual prompts $\{P_k^v\}_{k=1}^K$ tend to be more similar, whereas the textual prompts $\{P_t^v\}_{k=1}^K$ exhibit greater variability. To validate this conjecture, we measure the differences in both textual and visual prompts across all clients using drift diversity (Li et al., 2023a) and cosine distance, which respectively reflect the diversity in the amount and direction of prompts among clients. As illustrated in Figure 3, the differences in textual prompts are significantly greater than those found in visual prompts.

Thus, we propose a partial prompt aggregation protocol, where all clients upload their visual prompts $\{P_k^v\}_{k=1}^K$ to the server for aggregation while keeping their textual prompts $\{P_k^t\}_{k=1}^K$ local for personalization. For the server, we design the visual prompt aggregation strategy that utilizes a weighted average approach as follows:

$$\widetilde{P}^v = \sum_{k=1}^K \frac{\widetilde{n}_k}{\sum_i \widetilde{n}_i} \cdot P_k^v, \tag{4}$$

where $\widetilde{n}_k = \sum_c \widetilde{u}_{k,c}$ represents the total number of samples assigned to client $k$. The weighting mechanism ensures that clients with a larger allocated sample capacity have a greater influence on the aggregation process. The aggregated visual prompts $\widetilde{P}^v$ are then distributed back to the clients as initialization for their local models for the next training round. The overall objective function of FedPP is formulated as follows:

$$\min_{\widetilde{P}^v, \{P_k^t\}_{k=1}^K} \sum_{k=1}^K \mathbb{E}_{(x,\hat{y}) \in \widetilde{D}_k} \ell_{ce}\left(g(\widetilde{P}^v, P_k^t; x), \hat{y}\right), \tag{5}$$

where $g(\widetilde{P}^v, P_k^t; \cdot)$ represents the model output, $\hat{y}$ denotes pseudo labels, and $\ell_{ce}(\cdot, \cdot)$ is the cross-entropy loss function.

This approach enhances global performance by aggregating visual prompts $\{P_k^v\}_{k=1}^K$ while allowing clients to utilize personalized textual prompts $\{P_k^t\}_{k=1}^K$ that align better with their specific data distributions. Moreover, in comparison to methods that aggregate prompts from both modalities, our approach can reduce communication overhead, making it more practical for environments where communication resources are limited or constrained.

## 4 EXPERIMENTS

### 4.1 SETUPS

**Datasets.** We evaluated the performance of our method on six public benchmark datasets characterized by varying types of label skew. Following previous research (Guo et al., 2023a; Li et al., 2024; Cui et al., 2024), we utilized four representative visual classification datasets: **DTD** (Cimpoi et al., 2014), **RESISC45** (Cheng et al., 2017), **UCF101** (Soomro, 2012), and **CUB** (Wah et al., 2011), along with two standard federated learning benchmark datasets: **CIFAR10** and **CIFAR100** (Krizhevsky et al., 2009). We partitioned each dataset into distinct training and test sets, which were subsequently divided into non-overlapping subsets for different clients. To construct sample sets with label skew, we followed the settings outlined in (Li et al., 2022b) and employed two prevalent forms of label skew: quantity-based and Dirichlet-based. In the quantity-based label skew, all training data is grouped by label and allocated into shards with imbalanced quantities. The parameter $s$ signifies the number of shards per client, regulating the level of label skew (Lee et al., 2022). In the Dirichlet-based label skew, clients receive samples for each class based on the Dirichlet distribution (Zhu et al., 2021). Here, the parameter $\beta$ controls the degree of label skew, with lower values indicating greater label skews.

**Baseline methods.** In our experiments, we compare FedPP, with two popular pseudo-label selection methods in central unsupervised learning and four supervised federated learning methods. Regarding pseudo-labeling selection, **CPL** (Zhang et al., 2024a) selects the most reliable samples based on confidence for each class, while **FPL** (Menghini et al., 2023) generates multiple pseudo-labels for each sample through selection at both the sample and category levels. Zero shot learning involves using the pre-trained CLIP model with a hand-crafted textual prompt template, such as "a photo of a [CLASS]," to predict on the test data. For supervised federated learning methods, **PromptFL** (Guo

Table 1: Accuracies (%) of experiments under two degrees of **Dirichlet-based** label skews. FPL (Menghini et al., 2023) and CPL (Zhang et al., 2024a) are adopted as baseline pseudo-labeling (**PL**) method. All results are averaged over 3 runs. **Bold** and underline represent the best and second-best results, respectively.

| Method | PL | DTD | | RESISC45 | | CUB | | UCF101 | | CIFAR10 | | CIFAR100 | |
|---|---|---|---|---|---|---|---|---|---|---|---|---|---|
| | | $\beta = 0.1$ | $\beta = 0.05$ | $\beta = 0.1$ | $\beta = 0.05$ | $\beta = 0.1$ | $\beta = 0.05$ | $\beta = 0.1$ | $\beta = 0.05$ | $\beta = 0.1$ | $\beta = 0.05$ | $\beta = 0.1$ | $\beta = 0.05$ |
| Zore shot | - | 43.24 | 43.24 | 54.51 | 54.51 | 51.28 | 51.28 | 61.00 | 61.00 | 68.90 | 68.90 | 64.17 | 64.17 |
| PromptFL | FPL | 45.79 | 44.62 | 59.76 | 58.05 | 47.29 | 46.16 | 64.39 | 62.96 | 68.07 | 65.77 | 63.26 | 62.56 |
| PromptFL | CPL | 44.84 | 46.32 | 62.52 | 60.99 | 48.72 | 48.86 | 63.86 | 64.97 | 70.22 | 70.60 | 66.04 | 65.14 |
| Promptprox | FPL | 45.15 | 43.88 | 59.36 | 58.59 | 47.04 | 47.01 | 62.91 | 61.27 | 68.14 | 66.53 | 62.77 | 64.12 |
| Promptprox | CPL | 43.51 | 45.85 | 59.99 | 62.35 | 49.25 | 48.65 | 64.55 | 63.94 | 71.45 | 70.71 | 65.86 | 66.15 |
| pFedPrompt | FPL | 44.56 | 46.19 | 65.95 | 60.52 | 48.48 | 44.42 | 64.37 | 64.18 | 68.20 | 70.73 | 65.08 | 65.83 |
| pFedPrompt | CPL | 44.22 | 47.59 | 61.76 | 66.79 | 47.23 | 50.97 | 65.59 | 65.44 | 71.16 | 69.26 | 65.63 | 67.86 |
| FedOPT | FPL | 30.46 | 35.89 | 50.15 | 45.39 | 46.04 | 46.38 | 55.71 | 57.86 | 51.73 | 52.36 | 47.51 | 47.50 |
| FedOPT | CPL | 36.85 | 33.31 | 41.10 | 39.75 | 42.94 | 43.73 | 58.34 | 56.26 | 48.10 | 47.32 | 57.87 | 59.81 |
| **Ours** | **CPS** | **60.89** | **66.37** | **75.76** | **80.26** | **56.09** | **54.80** | **73.20** | **74.97** | **75.88** | **76.17** | **73.59** | **72.84** |

Table 2: Accuracies (%) of experiments under **quantity-based** label skews.

| Method | PL | DTD | RESISC45 | CUB | UCF101 | CIFAR10 | CIFAR100 |
|---|---|---|---|---|---|---|---|
| Zore shot | - | 43.24 | 54.51 | 51.28 | 61.00 | 68.90 | 64.17 |
| PromptFL | FPL | 43.53 | 57.53 | 46.66 | 62.35 | 70.38 | 63.94 |
| PromptFL | CPL | 43.82 | 61.11 | 47.31 | 64.23 | 70.74 | 67.23 |
| Promptprox | FPL | 44.89 | 56.71 | 48.74 | 61.80 | 67.96 | 64.01 |
| Promptprox | CPL | 45.53 | 60.77 | 47.61 | 63.86 | 70.99 | 66.49 |
| pFedPromp | FPL | 45.09 | 60.46 | 47.71 | 65.39 | 68.10 | 62.82 |
| pFedPrompt | CPL | 45.14 | 66.68 | 50.45 | 65.57 | 68.79 | 65.45 |
| FedOPT | FPL | 36.65 | 49.04 | 42.61 | 52.95 | 49.22 | 46.76 |
| FedOPT | CPL | 35.32 | 42.08 | 36.26 | 50.05 | 44.21 | 57.98 |
| **Ours** | **CPS** | **56.18** | **81.06** | **56.31** | **72.03** | **75.06** | **73.39** |

et al., 2023b) utilizes unified prompt vectors learned across clients via FedAvg (McMahan et al., 2017). **Promptprox**, introduced in Guo et al. (2023b), is derived from traditional federated learning technique FedProx (Li et al., 2020). **pFedPrompt** (Guo et al., 2023a) learns a unified prompt with personalized attention modules for local visual embeddings. Finally, **FedOPT** (Li et al., 2024) performs optimal transport between the global and local textual prompts to obtain an optimal model.

**Implementation details.** We adopt the widely recognized vision-language model CLIP ViT-B32 as our base model. We set the number of clients, $K$, to 5 for the CUB and UCF101 datasets, and to 10 for the other datasets, implementing full client participation by default. We conduct 20 communication rounds for all experimental datasets, regenerating pseudo-labels using the updated local model every 5 communication intervals. Within each communication round, local training spans 10 epochs. We optimize the prompts using mini-batch Stochastic Gradient Descent (SGD) with a learning rate of 0.1, a momentum of 0.9, and decay following the cosine annealing rule. For our proposed method, we set the global number of pseudo-labels for each class configured to one-quarter of the total data from all clients. Following the approach in FedOPT (Li et al., 2024), we conduct three trials for each experimental setting and report the mean accuracy. All experiments are conducted using PyTorch (Paszke et al., 2019) on NVIDIA 3090 GPUs. More details on the datasets and implementation can be found in the technical appendix A.2.

## 4.2 EXPERIMENTAL RESULTS

**Results under Dirichlet-based label skews with various datasets.** Table 1 presents the performance results of various methods with different levels of Dirichlet-based label skews ($\beta \in \{0.1, 0.05\}$). Our method, FedPP, significantly outperforms state-of-the-art algorithms across all datasets, confirming the effectiveness of our cooperative pseudo-label selection strategy and partial prompt aggregation protocol. Notably, FedOPT, the latest personalized supervised federated prompt tuning method, demonstrates that the results obtained when combining it with baseline pseudo-label selection methods are inferior to those of zero-shot learning. This is attributed to the low accuracy of the pseudo-labels chosen by each client, which do not accurately represent the true distribution of local data, a situation exacerbated by personalized training. We will discuss this further in the

Table 3: Accuracies (%) of combining proposed pseudo-labeling strategy CPS with existing feder-ated training methods under **Dirichlet-based** label skews ($\beta = 0.1$).

| Method | PL | DTD | RESISC45 | CUB | UCF101 |
|---|---|---|---|---|---|
| PromptFL | FPL | 45.79 | 59.76 | 47.29 | 64.39 |
| PromptFL | **CPS** | 47.34 (+1.55) | 61.68(+1.92) | 49.08(+1.79) | 64.97(+0.58) |
| pFedPrompt | FPL | 44.56 | 65.95 | 48.48 | 64.37 |
| pFedPrompt | **CPS** | 49.25(+4.69) | 67.97(+2.02) | 52.77(+2.29) | 67.93(+3.56) |
| FedOPT | FPL | 30.46 | 50.15 | 46.04 | 55.71 |
| FedOPT | **CPS** | 57.93(+27.47) | 74.85(+24.70) | 55.72(+9.68) | 69.21(+13.50) |

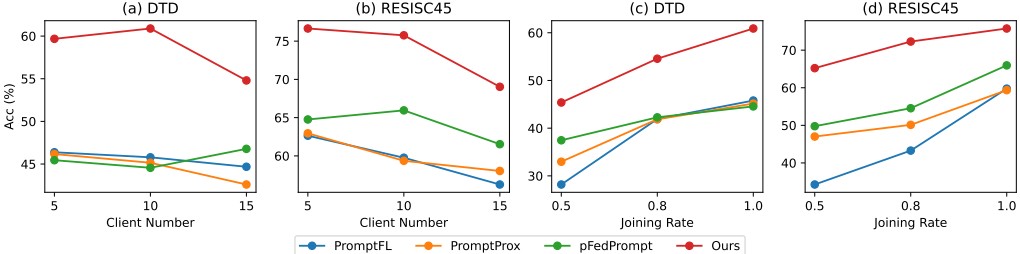

Figure 4: Results of experiments with various client numbers and different client joining rates under Dirichlet-based label skews ($\beta = 0.1$).

following sections. In the CUB dataset, all baseline results are worse than the zero-shot results. In contrast, our method consistently maintains excellent performance, underscoring the importance of the cooperative pseudo-label selection strategy and partial prompt aggregation protocol.

**Results under quantity-based label skews with various datasets.** We present the performance of all methods under quantity-based label distribution skews in Table 2, utilizing parameters $s = C \times 0.2$ for all datasets, where $C$ represents the number of classes in each dataset. In this setting, each client possesses samples from only a few classes, complicating pseudo-labeling and model training. Owing to this challenge, many baseline results are outperformed by the zero-shot approach. In contrast, our method maintains strong performance, similar to that observed with Dirichlet-based distributions, further underscoring the superiority of our approach.

### 4.3 ANALYSIS

**Comparation of different pseudo labels selection methods.** During the unsupervised training process using pseudo-labels, the selection of pseudo-labels directly determines the final model performance. Here, we compare our pseudo-label selection method with two baseline methods, CPL (Zhang et al., 2024a) and FPL (Menghini et al., 2023). As shown in Table 1 and Table 2, for the same federated training method, CPL outperforms FPL on datasets such as DTD and CUB, whereas the opposite is true for the CIFAR datasets. In contrast, our cooperative pseudo-label se-lection strategy consistently outperforms both baseline methods across all datasets. This is due to the significantly higher accuracy of the pseudo-labels generated by our selection method compared to the others. To further demonstrate the effectiveness of our approach, we combine our pseudo-label selection method with other state-of-the-art supervised federated training methods. As shown in Table 3, our method significantly improves the performance of baseline pseudo-label selection methods across different datasets, with a notable increase of up to 27.47% on the DTD dataset. Overall, these results strongly demonstrate the effectiveness of our pseudo-label selection method.

**Results under different client numbers.** We analyze our proposed method's performance against baseline methods across varying numbers of clients. Unless specified otherwise, our experiments focus on Dirichlet-based skews with the parameter $\beta = 0.1$. We divide the DTD and RESISC45 datasets into 5, 10, and 15 clients, showcasing their final accuracy in Figure 4 (a) and (b). Re-markably, our method consistently outperforms the baseline methods, regardless of the number of clients. This demonstrates the robustness and scalability of our approach, ensuring efficient and effective learning even as the number of participating clients fluctuates.

Table 4: Accuracies (%) of experiments using **CLIP ViT-B16** as base model under Dirichlet-based label skews ($\beta = 0.1$) across four benchmarks.

| Method | PL | DTD | RESISC45 | CUB | UCF101 |
|---|---|---|---|---|---|
| Zero shot | - | 42.87 | 56.61 | 55.16 | 65.13 |
| PromptFL | FPL | 44.36 | 61.36 | 51.95 | 64.60 |
| PromptProx | FPL | 47.12 | 61.01 | 49.44 | 65.31 |
| pFedPrompt | FPL | 47.65 | 63.18 | 50.19 | 65.19 |
| FedOPT | FPL | 39.52 | 48.94 | 49.67 | 61.68 |
| Ours | **CPS** | 56.83 | 77.50 | 59.78 | 78.29 |

Table 5: **Ablation study**. Accuracies (%) under Dirichlet-based label skews. Conf. and Ent. denote confidence-based and entropy-based filters. G.A. represents global allocation.

| Conf. | Ent. | G.A. | PPA | DTD | RESISC45 | CUB | UCF101 |
|---|---|---|---|---|---|---|---|
| - | - | - | - | 45.79 | 59.76 | 47.29 | 64.39 |
| - | - | - | ✓ | 34.13 | 42.70 | 44.23 | 54.63 |
| - | - | ✓ | ✓ | 46.35 | 69.60 | 51.98 | 65.31 |
| - | ✓ | ✓ | ✓ | 58.83 | 72.28 | 54.91 | 70.23 |
| ✓ | - | ✓ | ✓ | 55.59 | 73.66 | 50.38 | 69.13 |
| ✓ | ✓ | ✓ | - | 47.34 | 61.68 | 49.08 | 64.97 |
| ✓ | ✓ | ✓ | ✓ | 60.89 | 75.76 | 56.09 | 73.20 |

**Impact of client joining rates.** In this analysis, we investigate variations in participation rates, considering values from $\{0.5, 0.8, 1.0\}$. As illustrated in Figure 4 (c) and (d), our method consistently outperforms other approaches across all participation rates. As the client participation rate decreases, the accuracy of all methods declines significantly. This instability is expected, as a lower client participation rate amplifies the divergence between randomly participating clients and the global model, resulting in erratic convergence. However, our method remains the best performer, highlighting its robustness to varying participation rates.

**Results under different image encoder backbone.** We further conduct experiments to evaluate the effect of different image encoders. The comparison results on DTD, RESISC45, CUB, and UCF101 datasets using ViT-B16 are presented in Table 4. Our method consistently surpasses previous approaches, demonstrating the effectiveness of our strategy in enhancing the performance of CLIP in unsupervised federated prompt tuning when smaller image encoders are employed. These experiments underscore the versatility and robustness of FedPP in real-world federated learning scenarios utilizing various backbone architectures.

**Effectiveness of each component.** Our approach comprises two key modules: a cooperative pseudo-label selection strategy (CPS) and a partial prompt aggregation (PPA) protocol. The results presented in Table 5 reveal that the two filtering criteria (confidence and entropy) and global allocation in the cooperative pseudo-label selection strategy contribute to significant performance improvements compared to the pseudo-label selection method FPL (Menghini et al., 2023). Similar to the results of FedOPT in Tables 1 and 2, without the cooperative pseudo-label selection strategy, our personalized training results are inferior to those of zero-shot learning. This is due to the fact that, without suitable pseudo-labels, the personalized strategy exacerbates clients' misclassification of unlabeled data. Comparing the last and second-to-last rows of Table 5 illustrates that our personalized method can achieve significant improvements under appropriate pseudo-label conditions. These findings demonstrate the effectiveness of our two key modules in improving overall model performance in federated learning scenarios with label skews.

## 5 CONCLUSION

In this paper, we introduce an unsupervised federated learning problem with CLIP, where clients with unlabeled data employ collaborative training for better performance without data sharing. To address such problem, we propose FedPP, an unsupervised solution including a cooperative pseudo labels selection strategy and a partial prompt aggregation protocol. The pseudo labels selection strategy allows the server to customize the selection process for each client, taking into account both local and global pseudo labels distributions. The aggregation protocol only aggregates visual prompts on the server for performance improvements through collaboration among clients and textual prompts are kept locally for better personalization by each client. Extensive results demonstrate the effectiveness of both components, and proposed FedPP outpergorms baseline methods across diverse datasets and various degrees of label skews. In future work, we will conduct a theoretical analysis of FedPP, including convergence, privacy, fairness, and other pertinent considerations.

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

# A APPENDIX

## A.1 THE PSEUDOCODE OF OUR METHOD

Here, we provide detailed descriptions of the algorithm for our FedPP, as shown in Algorithm 1. For the pseudo labels selection, each client uploads their estimated distribution $U_k^{est}$, which is filtered by confidence and entropy, to the server for cooperative pseudo labels assignment. The server globally selects $M$ pseudo labels for each category and allocates them to clients according to their estimated distribution, where the training data distribution for client $k$ is $\widetilde{U}_k$. Finally, client k constructs the training set $\widetilde{D}_k$ by selecting the most confident samples according to the capacity $\widetilde{U}_k$. For personalization, we only aggregate visual prompts on the server while keeping textual prompts locally.

---

**Algorithm 1 FedPP**

---

**Input:** number of communication rounds $T$, number of clients $K$, unlabeled dataset $\{D_k\}_{k=1}^K$, client participating rate $R$, number of local epochs $E$, batch size $B$, learning rate $\eta$, pseudo labels update interval $Q$.

**Output:** the global visual prompt $P^v$ and personalized textual prompts $\{P_k^t\}_{k=1}^K$

1: initialize $P^{v,0}$, $\{P_k^{t,0}\}_{k=1}^K$
2: $m \leftarrow \max(\lfloor R \cdot K \rfloor, 1)$
3: **for** *communication round* $r = 1, 2, \cdots, T$ **do**
4:    **if** r % Q = 0 **then**
5:       $\{\widetilde{D}_k\}$ = **pseudo labels_Selection**$(\{D_k\}_{k=1}^K)$
6:    **end if**
7:    $M_r \leftarrow$ randomly select a subset containing $m$ clients
8:    **for** *each client* $k \in M_t$ **do**
9:       $P_k^{v,r} = P^{v,r}$, $P_k^{t,r} = P_k^{t,r}$
10:      $P_k^{v,r+1}, P_k^{t,r+1} \leftarrow$ **LocalUpdate**$(P_k^{v,r}, P_k^{t,r})$
11:    **end for**
12:    $P^{v,r+1} = P^{v,r} + \sum_{k \in M_r} \frac{|\widetilde{D}_k|}{|\widetilde{D}|}(P_k^{v,r+1} - P_k^{v,r})$
13: **end for**

14: **pseudo labels_Selection**$(\{D_k\}_{k=1}^K)$:
15: **for** $k = 1, ..., K$ **do**
16:    $D_k^{est} = \{(x, \hat{y})| \max_c p_c(x) > p_\alpha, Ent(p_c(x)) < H_\alpha\}$, Eq2
17:    $\widetilde{U}_k = \{\widetilde{u}_{k,1}, \widetilde{u}_{k,2}, ..., \widetilde{u}_{k,C}\}, \widetilde{u}_{k,c} = \frac{u_{k,c}}{\sum_{k=1}^K u_{k,c}} \times M$, $u_{k,c}$ is the number of data for class $c$ in
     $D_k^{est}$, Eq3
18:    $\widetilde{D}_k$ is selected by the most confident samples according to the capacity $\widetilde{U}_k$
19: **end for**
20: **return** $\{\widetilde{D}_k\}_{k=1}^K$

21: **LocalUpdate**$(P_k^{v,r}, P_k^{t,r})$:
22: **for** *epoch* $e = 1, 2, \cdots, E$ **do**
23:    **for** *each batch* $\mathcal{B}_i = \{x, y\} \in \widetilde{D}_i$ **do**
24:       $\mathcal{L}(P^v, P^t; \mathcal{B}_i) = -\mathbb{E}_{(x,y) \sim \mathcal{B}_i} \log \left( \frac{e^{f(P^v, P^t; x)[y]}}{\sum_c e^{f(P^v, P^t; x)[c]}} \right)$
25:       $P_k^{v,r} = P_k^{v,r} - \eta \nabla \mathcal{L}(P_k^{v,r}; \mathcal{B}_i)$
26:       $P_k^{t,r} = P_k^{t,r} - \eta \nabla \mathcal{L}(P_k^{t,r}; \mathcal{B}_i)$
27:    **end for**
28: **end for**
29: **return** $P_k^{v,r}, P_k^{t,r}$

---

## A.2 EXPERIMENTAL DETAILS

**Details of Dataset Setup.** For our evaluation, we have chosen six diverse visual classification datasets as our benchmark. The detailed statistics of each dataset are shown in Table 6, including the original tasks, the number of classes, the size of training, and testing samples.

Table 6: The detailed statistics of datasets used in experiments.

| Dataset | Task | Classes | Training Size | Testing Size |
|---|---|---|---|---|
| CUB | Image classification | 200 | 5,594 | 5,794 |
| RESISC45 | Scene classification | 45 | 6,300 | 25,200 |
| UCF101 | Action recognition | 101 | 7,639 | 3,783 |
| DTD | Texture recognition | 47 | 2,820 | 1,692 |
| CIFAR10 | Image classification | 10 | 50,000 | 10,000 |
| CIFAR100 | Image classification | 100 | 50,000 | 10,000 |

We employed quantity-based and Dirichlet-based label skews to construct data heterogeneity. For quantity-based distribution, each client has category counts of 10, 58, 66, 30, 2, and 20 in the DTD, RESICS45, CUB, UCF101, CIFAR10, and CIFAR100 datasets, respectively. For Dirichlet-based label skews, we used $\beta = \{0.1, 0.05\}$ to generate data for each client. Here, we present the data distribution for each case, using CIFAR-10 as an example.

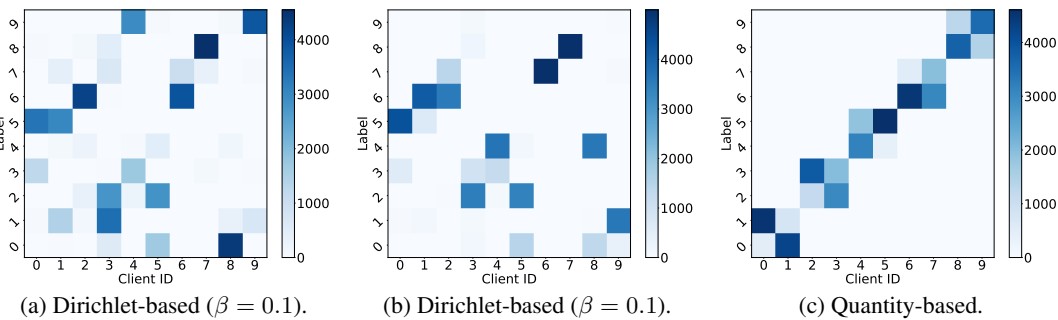

(a) Dirichlet-based ($\beta = 0.1$).    (b) Dirichlet-based ($\beta = 0.1$).    (c) Quantity-based.

Figure 5: (a) and (b) depict label skews of Dirichlet-based label skews and (c) presents the quantity-based label skew.

**Implementation details.** All input images across datasets are resized to 224 × 224 pixels and further divided into 14 × 14 patches with a dimension of 768. We take deep visual prompts into our implementation and we add trainable prompts with a dimension of $5 \times 867$ to the output of each transformer layer in the visual encoder. For text prompts, we set the length to 16 with a dimension of 512. Batch sizes are set to 64 for both training and testing.

## A.3 DRIFT DIVERSITY

Following(Li et al., 2023a), we employ drift diversity to assess magnitude differences. Specifically, drift diversity is defined as follows:

$$\xi^r := \frac{\sum_{k=1}^{K} \|m_k^r\|^2}{\|\sum_{k=1}^{K} m_k^r\|^2} \quad \text{with} \quad m_k^r = P_k^r - P^{r-1} \tag{6}$$

where $P_k^r$ is updated prompt of client $k$ in round $r$ and $P^{r-1}$ is aggregated prompt on the server in round $r - 1$.

## A.4 ADDITIONAL EXPERIMENTS RESULTS

**Comparison of pseudo-label accuracy.** As shown in Table 7, we present the accuracy of different pseudo-label selection methods, utilizing CLIP's zero-shot prediction results. Obviously, our pseudo

Table 7: Pseudo-label accuracy of different method with Dirichlet-based label skews ($\beta = 0.1$) on various datasets. The baseline pseudo-label selection method is FPL (Menghini et al., 2023) and CPL (Zhang et al., 2024a).

|          | DTD   | RESISC45 | CUB   | UCF101 | CIFAR10 | CIFAR100 |
|----------|-------|----------|-------|--------|---------|----------|
| CPL      | 31.84 | 43.04    | 41.04 | 45.72  | 41.18   | 39.05    |
| FPL      | 34.32 | 40.42    | 47.75 | 50.57  | 37.02   | 57.81    |
| **Ours** | 78.74 | 84.73    | 89.30 | 85.12  | 87.30   | 86.13    |

labels selection method significantly improves accuracy against baseline pseudo labels selection methods across different datasets.

