# OpenReview forum: "Personalized Prompt Tuning for Unsupervised Federated Learning"
_ICLR.cc/2025/Conference — ICLR 2025 Conference Withdrawn Submission_

### Official Review · Reviewer_7gDR · 2024-10-31

**Soundness:** 2
**Presentation:** 2
**Contribution:** 2
**Rating:** 3
**Confidence:** 4

**Summary:**

Federated learning struggles with unsupervised classification due to the lack of category knowledge. Leveraging CLIP's zero-shot classification capabilities, this paper introduces FedPP for unsupervised federated learning with unlabeled heterogeneous data. FedPP employs a cooperative pseudo-label selection strategy for balanced class training and a partial prompt aggregation protocol to optimize global performance while supporting local personalization. Extensive experiments on six datasets demonstrate the effectiveness of FedPP in enhancing global performance.

**Strengths:**

1.  The proposed FedPP method addresses a key challenge, unlabeled heterogeneous data in FL, with its cooperative pseudo-label selection and partial prompt aggregation.
2. By utilizing CLIP's zero-shot capabilities, FedPP reduces reliance on labeled data, which is a significant advantage for unsupervised FL scenarios.
3. The paper conducts extensive experiments across six datasets, demonstrating the effectiveness of FedPP.

**Weaknesses:**

1. In line 68, the paper states that "textual prompts tend to learn category-related information which may introduce conflicts due to label skews." Could the authors provide references to any existing literature that discusses this problem or present simple results to illustrate this statement?

2. In FedPP, clients are required to deploy CLIP models, specifically ViT-B32 in the experimental settings. This model may be too large for some clients to deploy. Could the authors provide results using a smaller model, such as ResNet50, for CLIP?

3. The connection between the Partial Prompt Aggregation (PPA) and Cooperative Pseudo-label Selection (CPS) methods appears to be weak. I did not observe any clear connections between these two methods. Could the authors clarify the motivations and connections between Sections 3.3 and 3.4 more clearly?

4. In line 242, how to determine $M$ in the server? Is this parameter manually decided, and does it vary for different datasets?

5. In line 270, the paper states that "the textual branch focuses on determining the classification boundaries by leveraging category information." However, from Figure 3, it seems to suggest that "textual prompts are more personalized than visual prompts in CLIP." Figure 3 does not appear to illustrate any information about different categories. Could the authors clarify this point?

6. In the experiments, do all baselines use CLIP as their client models?

7. The number of clients in the experiments is relatively small, with only 5 or 10 clients. Is it possible to extend the experiments with a larger number of clients (like 50 or 100 clients) using smaller client models (smaller ViT or ResNet)?

**Questions:**

Please see the weaknesses.

---

### Official Review · Reviewer_CYM7 · 2024-11-02

**Soundness:** 2
**Presentation:** 2
**Contribution:** 2
**Rating:** 5
**Confidence:** 3

**Summary:**

This work proposes a unsupervised federated learning approach CPS based on pseudo-labeling method and aided by two key components: (1) cooperative pseudo labels selection; (2) partial prompt aggregation. Specifically, client filters samples from the unlabeled local dataset based on confidence and entropy and sends the etimated labeling distribution to the server for aggregation. The aggregated global  pseudo-labeling distribution is sent back to the server for ensuring uniform training across all categories. Partial prompt aggregation protocol allows the clients only to upload visual prompts to the server and keep their textual prompt for personalization. Experimental results demonstrate the improvement of CPS.

**Strengths:**

1. The paper attempted to solve an importance issue: unsupervised learning under federated settings.
2. The paper conducts experiments on six benchmarks and the proposed scheme consistently performs well.
3. Introducing CLIP model in federated settings is interesting and might have good prospects.

**Weaknesses:**

1. In the some federated learning settings, such as on-device federated learning, it is unrealistic to assume clients are able to run CLIP model which requires much more computational resources.
2. There is no clear insight for the partial prompt aggregation protocol. Why only uploading visual prompts are benefitical to the performance. It should be explained more clearly.
3. The number of clients in the experiments are too small. It would be helpful for the paper to have experiments with more than 100 clients.

**Questions:**

1.The problem formulation in section 3.2 is very confusing. It claimed that each client process an unlabeled local dataset and then the challege of the problem is stated as label skew.  It is not clear whether the clients have local datasets with labels or not.

---

### Official Review · Reviewer_RTg4 · 2024-11-02

**Soundness:** 3
**Presentation:** 3
**Contribution:** 2
**Rating:** 5
**Confidence:** 2

**Summary:**

This paper explores unsupervised federated learning using the CLIP model, where clients collaborate on tasks like image classification without sharing labeled data. To address challenges such as category bias and label skews, the authors propose a method called FedPP. This method includes: a) Cooperative pseudo-label selection, which ensures balanced pseudo-label distribution across all categories, and b) Partial prompt aggregation, where only visual prompts are shared to improve both personalization and global performance. The effectiveness of the proposed method was validated across multiple datasets with different types of heterogeneity.

**Strengths:**

1. The paper is clearly structured, and the content is easy to understand.
2. Authors have provided both pseudocodes and diagrams to present the proposed method in detail.
3. Authors have provided extensive experimental data in the paper, evaluating the performance of the proposed FedPP method from multiple perspectives.
4. FedPP demonstrates strong performance, showing significant improvements over baseline methods across multiple datasets and scenarios.
5. According to the paper, the two main strategies CPS (Cooperative pseudo-label selection) and PPA (Partial prompt aggregation) are not only easy to implement but also work well in combination with other training algorithms.

**Weaknesses:**

1. The main limitation of the paper is the lack of theoretical analysis. FedPP introduces significant differences in training compared to previous baselines, which could impact training convergence, communication complexity, and privacy concerns. However, none of these issues are discussed in detail. Specifically, the following points require further clarification:
a) Communication Cost: In FedPP, clients incur an additional cost by uploading the estimated pseudo-label distribution $U_k^{est}$, while avoiding the need to upload textual prompts since only visual prompts are aggregated by the server. How do these changes impact the overall communication cost? From a complexity perspective, is FedPP more or less efficient than other methods?
b) Training Convergence: If clients update both visual and textual prompts locally but only visual prompts are aggregated by the server, can FedPP guarantee convergence? How does the convergence speed of FedPP compare to that of other methods?
c) Privacy Concerns: The uploaded pseudo-label distribution $U_k^{est}$ could potentially reveal statistics about local data, as it is derived from high-confidence samples. This raises concerns that FedPP may pose a higher risk to client privacy compared to traditional federated learning methods, which only share model weights. Is there theoretical evidence to demonstrate that FedPP does not compromise client privacy?
2. The paper highlights the two modules in FedPP as key contributions. However, the Partial Prompt Aggregation (PPA) module can be simply summarized as aggregating only the visual part of the prompts on the server, and the aggregation method used in FedPP is essentially the same as prior federated approaches that rely on local data capacity for aggregation. Limited novelty and innovation is shown with this module. Furthermore, although Figure 3 demonstrates that the experimental divergence on textual prompt is more significant than that on visual prompt, it remains under-explored why aggregating only visual prompts leads to performance improvement compared to previous methods that tend to aggregate both prompts. Here some in-depth exploration to justify this aggregation approach is important for understanding but unfortunately is not provided.
3. The paper also contains numerous mathematical notation and spelling errors, impacting the readability to some extent. For example, in Equations (3) and (4), the symbol $i$ is used without explanation. It seems to represent client indexes, but the paper mentions that this is denoted by $k$. Additionally, in Tables 1 and 2, "zero shot" is misspelled as "zore shot." Furthermore, in appendix, there are also several symbol errors in Algorithm 1, such as $k \in M_t$ on line 9 and the inconsistent use of $p_\alpha$ and H_\alpha$ on line 16. Equation 2 shows $p_{1/2}$ and $H_{1/2} instead.

**Questions:**

Please respond to the weaknesses.

---

### Official Review · Reviewer_pxP5 · 2024-11-04

**Soundness:** 2
**Presentation:** 3
**Contribution:** 2
**Rating:** 3
**Confidence:** 4

**Summary:**

The paper proposes *FedPP* to address the problem of unsupervised federated learning (FL) for classification tasks using ContrastiveLanguage-ImagePre-training (CLIP). FedPP method uses a cooperative pseudo-label selection strategy to balance training across all classes and a partial prompt aggregation protocol to enhance both global and personalized performance. Experimental results on six datasets show that FedPP outperforms baseline methods by 5-15% in prediction accuracy, particularly in handling label skew across clients.

**Strengths:**

++ The paper has a clear workflow and demonstration.

**Weaknesses:**

-- Overclaimed statements about CLIP and FedPP. See C1.

-- The selection strategy for reliable samples needs further justification. See C2.

-- Generating pseudo-labels based on 'reliable' samples may not fit the local data distributions. See C3.

-- The paper lacks a corresponding security design for transmitting information between clients and the server. See C4.

**Questions:**

C1:
  - Line 049-050: CLIP’s zero-shot capabilities are powerful for classification tasks without labeled data, as it can use pre-trained image-text embeddings to generate predictions across various classes. However, in complex federated settings with non-independent and identically distributed (non-IID) data across clients, challenges like label skew, domain heterogeneity, and client-specific needs can still impact performance. Hence, the paper cannot use the word 'eliminate.'
  - Line 077-080: The paper primarily focuses on image classification problems under unsupervised FL. Please refrain from claiming the problem to address is the unsupervised federated learning problem, which is a more prominent topic.

C2:
  - This paper selects reliable samples based on the median of confidence and entropy within the alternative set without any explanation of the alternative set. Hence, the paper must explain the meaning of the alternative set.
  - Confidence-based filtering may include confidently incorrect predictions, especially if the model is biased towards certain classes due to imbalanced data. This can introduce pseudo-label errors, particularly in non-IID settings where client data distributions differ significantly. Hence, the paper must justify why the alternative set could help FedPP select reliable samples.
  - The selection strategy of reliable samples is based on two thresholds, p_{1/2} and H_{1/2}. Hence, the paper needs to justify why such thresholds generalize across all clients or empirically prove the generalization of such threshold values.

C3:
  - Since pseudo-labels are generated based on CLIP's initial zero-shot predictions, they may need to be more accurate, especially in domain shifts or non-standardized data distributions across clients. However, reliable samples only mean the corresponding prediction labels are correct with high probability. Hence, even if reliable samples are picked correctly, the paper still needs to justify why selected reliable samples represent the local data distribution.

C4:
  - If the estimated pseudo-labeling distribution is highly similar to the true label distribution as expected, uploading such information to the server introduces additional security concerns since threat models may utilize such information to conduct attacks like data distribution inference attacks. Hence, the paper must justify that the system design would not lead to privacy issues.

**Writing Issues**

  1. Abstract： The abbreviation 'CLIP' is not explained. According to the citation, it is 'ContrastiveLanguage-ImagePre-training.'
  2. Line 207: The paper should briefly explain reliable samples.

---

### Note · Authors · 2024-11-15

I have read and agree with the venue's withdrawal policy on behalf of myself and my co-authors.